# Donor, Recipient and Surgeon Sex and Sex-Concordance and their Impact on Liver Transplant Outcome

**DOI:** 10.3390/jpm13020281

**Published:** 2023-02-01

**Authors:** Laura Ioana Mazilescu, Isabel Bernheim, Jürgen Treckmann, Sonia Radunz

**Affiliations:** Department of General, Visceral and Transplant Surgery, University Hospital Essen, 45147 Essen, Germany

**Keywords:** females, graft survival, liver transplantation, patient survival, sex, sex-concordance

## Abstract

(1) Background: Patient sex is associated with differential outcome of many procedures although the exact mechanisms remain unknown. Especially in transplant surgery, surgeon-patient sex-concordance is rarely present for female patients and outcome may be negatively affected. (2) Methods: In this single-center retrospective cohort study, recipient, donor, and surgeon sex were evaluated and short- and long-term outcome was analyzed with regards to sex and sex-concordance of patients, donors, and surgeons. (3) Results: We included 425 recipients in our study; 50.1% of organ donors, 32.7% of recipients, and 13.9% of surgeons were female. Recipient-donor sex concordance was present in 82.7% of female recipients and in 65.7% of male recipients (*p* = 0.0002). Recipient-surgeon sex concordance was present in 11.5% of female recipients and in 85.0% of male recipients (*p* < 0.0001). Five-year patient survival was comparable between female and male recipients (70.0% vs. 73.3%, *p* = 0.3978). Five-year patient survival of female recipients treated by female surgeons was improved without reaching significance (81.3% vs. 68.4%, *p* = 0.3621). (4) Conclusions: Female recipients and female surgeons are underrepresented in liver transplant surgery. Societal factors influencing outcome of female patients suffering from end-stage organ failure need to be further examined and acted upon to possibly improve the outcome of female liver transplant recipients.

## 1. Introduction

Recently, a negative association of patient-surgeon sex-discordance with surgical outcomes following common surgical procedures was demonstrated. Outcomes were especially worse among female patients treated by male surgeons [1]. Furthermore, a decreased 30-day mortality risk for patients treated by female surgeons was reported [2]. Nevertheless, further psychological/behavioral factors, i.e., patients’ trust or distrust in surgeons and hospitals, may also influence outcomes of surgical treatments.

Non-surgical patients treated in hospital showed lower 30-day mortality and readmission rates if treated by a female general internist [3]. In primary care, gender disagreement was found to lead to diminished compliance, decreased certainty of diagnosis and lower accuracy in diagnosing a severe condition [4]. Patient-provider sex concordance positively affects the compliance of patients in attending routine screening for colorectal, cervical and breast cancer [5]. In patients with acute myocardial infarction, physician-patient gender concordance seems to affect the mortality rate, with women less likely to survive when treated by a male doctor [6]. In addition, gender concordance is associated with better agreement on advice on nutrition, exercise and weight loss [7]. Studies suggest that female doctors are more prone to use a patient focused approach, while male doctors are less likely to use evidence-based guidelines [8,9].

Gender diversity in medicine has improved in the last decades, however in the field of surgery a persistent gender disparity remains [10]. Studies from England and the United States report that less than 25% of doctors in most surgical specialties are women [11] and less than 10% of the chairs of surgical departments are female [12].

Gender inequality on the transplant list represents another challenge in liver transplantation [13,14]. Women awaiting liver transplantation show higher rates of waitlist mortality than men and studies have shown that MELD score underestimates disease severity in women with a significant impact on allocation [15]. Several hypotheses have been proposed to explain the higher waitlist mortality in women, such as underestimation of renal function impairment and the physical stature of women [16,17,18]. However, it remains uncertain, whether other factors might also explain the sex differences in waitlist outcomes.

The extent at which sex differences affect access to transplant, graft outcomes and patient survival are multifactorial [14,19,20]. Sex differences could vary by age. The expected sex differences in mortality rates should be considered when interpreting survival rates between women and men. To give patients an equal chance to be listed for transplantation and to receive a suitable organ more studies need to further investigate the effect of sex and gender in transplantation.

With women also being underrepresented in the transplant field [21,22], surgeon-patient discordance should be rather high in liver transplant surgery. Hence, objective of this single-center retrospective cohort study was to analyze the presence of sex discordance and its impact on short- and long-term outcome of liver transplantation.

## 2. Materials and Methods

In this single-center retrospective cohort study, all adult patients undergoing first-time deceased-donor liver transplantation between 2013 and 2018 at our transplant center were included. All patients received donation after brain death grafts. The study was approved by the local ethics committee (19-8729-BO) and was conducted in accordance with the Helsinki Declaration of 1975, as revised in 2008. Inclusion criteria were all adult, first-time liver transplant recipients. The requirement for informed consent was waived due to the retrospective design of the study.

Patients underwent orthotopic liver transplantation without veno-venous bypass when a suitable organ offer became available. In general, immunosuppression consisted of tacrolimus 0.1 mg/kg adjusted to a trough level of 5-7 ng/mL, mycophenolate mofetil 1 g bid and prednisolone 20 mg, tapered and withdrawn within six weeks. 

The following baseline characteristics of transplant recipients and organ donors were assessed: sex, age, body mass index (BMI), underlying disease resulting in liver failure, model of end-stage liver disease (MELD) score, cause of brain death, and donor risk index (DRI). Cold ischemia time (CIT) and warm ischemia time (WIT) were evaluated as well.

Short- and long-term outcome was analyzed with regards to sex and sex-concordance of patients, donors, and surgeons. Short-term outcome was defined as any complication within the initial hospital stay and was graded according to the Clavien-Dindo classification [16]. All complications grade 3 and higher were recorded. Long-term outcome was defined as 5-year-survival after liver transplantation.

All data were tested for normality using the D'Agostino&Pearson omnibus normality test. Categorical variables are presented as percentages and continuous variables as median (range), unless stated otherwise. Differences were tested using Fisher’s exact test or chi-square test, as appropriate. Differences in continuous variables were tested using student’s t-test or Mann-Whitney test, as appropriate. Patient and graft survival were evaluated using the Kaplan-Meier method and compared with the log-rank test. The reference point for all calculations of survival was the day of liver transplantation. Graft survival was determined until re-transplantation, death, or the end of the study period. A *p* value ≤ 0.05 (two-tailed) was considered to be significant. Cox regression analysis was performed to assess the relationship between survival time and covariates. Final variables incorporated in the model included recipient age, sex, MELD score, DRI, and patient-surgeon sex concordance. All analyses have to be regarded as exploratory as we did not adjust the significance level globally in terms of the multiple testing problem. 

Data collection and statistical analysis were performed using Microsoft Excel for Mac version 16.63.1 (Microsoft Corporation, Redmond, WA, USA) and GraphPad Prism version 9.4.1 for macOS (GraphPad Software, San Diego, CA, USA). Cox regression analysis was performed using IBM SPSS Statistics (version 23.0 for Windows, SPSS, Inc., Chicago, IL, USA).

## 3. Results

We included 425 recipients in our study. Organ donors were equally distributed according to sex (50.1% female, 49.9% male). The majority of the recipients were male (67.3%), and surgeons were predominantly male (86.1%).

Recipient, donor and surgical details stratified according to recipient sex are given in Table 1. Recipient age and MELD score were comparable between female and male recipients. Female recipients were of significantly less body weight (BMI 24.4 (15.1–41.5) vs. 26.2 (16.1–44.3) kg/m^2^, *p* = 0.0009), and underlying diseases resulting in the need for liver transplantation differed significantly between female and male recipients. DRI of organs transplanted into female recipients was significantly worse (1.837 (1.040–2.855) vs. 1.746 (0.9730–2.678), *p* = 0.0272). While CIT was comparable between female and male recipients, WIT was significantly shorter in female recipients (28 (16–65) vs.30 (16–80) minutes, *p* = 0.0009).

Donor characteristics stratified according to donor sex are given in Table 2. Female organ donors were of older age (60 (1–89) vs. 56 (1–87) yrs, *p* = 0.0182) and less body weight (BMI 25.6 (12.5–51.4) vs. 26.3 (13.3–58.8) kg/m^2^, *p* = 0.0012). Brain death in female donors was primarily due to cerebro-vascular accidents and less frequently due to trauma or anoxia. DRI was increased as compared to male donors (1.908 (0.9730–2.678) vs. 1.672 (0.9730–2.855), *p* < 0.0001).

Baseline recipient and donor characteristics were comparable between recipients treated by female and male surgeons, i.e. the complexity of the procedure was not different between female and male surgeons (Table 3). WIT was longer in recipients treated by female surgeons (34 (16–65) vs. 29 (16–80) min, *p* = 0.0002).

Sex-concordance with organ donors was more frequent in female recipients (82.7% vs. 65.7% in male recipients, *p* = 0.0002), while sex-concordance with surgeons was less frequent in female recipients (11.5% vs. 85% in male recipients, *p* < 0.0001).

Frequency of postoperative complications Dindo-Clavien classification grade 3 and higher was 42.1% in the entire study cohort. Complication rate was similar in female and male recipients (45.3% vs. 40.6%, *p* = 0.4023). Complications/interventions included necessity for dialysis (*n* = 2), surgical revision (*n* = 84), endoscopic retrograde cholangiopancreatography (*n* = 21), colonoscopy/endoscopy (*n* = 2), coronary angiography (*n* = 2), biopsy (*n* = 15) and death (*n* = 60). Recipients who received an organ from a female donor had a higher complication rate than recipients of organs from a male donor (48.8% vs 35.4%, *p* = 0.0059). Surgeon sex did not have an influence on the complication rate of the recipients (female surgeons 42.4% vs. male surgeons 42.1%, *p* > 0.9999).

Graft loss occurred in 6.5% (*n* = 9) of female recipients at day 8 (1–242) after initial liver transplantation and in 3.2% (*n* = 9) of male recipients at day 2 (2–1203) after initial liver transplantation (*p*=0.1262). Overall survival of recipients who had required re-transplantation was dismal; 44.4% of female re-transplant recipients and 66.7% of male re-transplant recipients died. Graft loss in the early phase after liver transplantation occurred due to primary non-function (*n* = 7, 2 (2–7) days post-LT) or hepatic artery thrombosis/dissection (*n* = 5, 6 (1–15) days post-LT). In five recipients, graft loss was due to biliary complications, that is ischemic-type biliary lesions (*n* = 4) or PSC recurrence (*n* = 1). These recipients underwent re-transplantation 242 (14–1203) days after initial liver transplantation. One patient developed severe cellular/humoral rejection that did not respond to pulsed dose prednisolone, thymoglobuline, and plasmapheresis. Therefore, this patient underwent successful re-transplantation 14 days after initial liver transplantation without further immunological problems.

Patient survival was comparable between female and male recipients (Figure 1). In female recipients, patient survival was 77.6% at one year and 70% at five years. In male recipients, patient survival was 82.1% at one year and 73.3% at five years. Neither in female nor in male recipients was five-year patient survival affected by recipient-donor sex-concordance (Figure 2). 

Female recipients treated by female surgeons demonstrated an improved five-year patient survival, yet this did not reach statistical significance (81.3% vs. 68.4%, *p* = 0.3621). In male recipients, 5-year survival was not affected by patient-surgeon sex-concordance (73.1% vs. 74.1%, *p* = 0.7238), see Figure 3. Comprehensive results of the cox regression analysis are depicted in Table 4; increasing recipient age and MELD score were independently associated with increased odds of post-transplant death.

## 4. Discussion

Liver transplantation represents an emergency surgical procedure where patients may rarely build a relationship with their preferred surgeon. Hence, confidence between patients and physicians should be of less importance for treatment outcomes. Furthermore, transplant waitlist patients suffer from a similar, life-threatening disease and undergo the same procedure; therefore, additional confounding factors are limited in this cohort.

In our center, one seems to suppose that the female sex is more likely to tolerate donor and recipient risk factors. Female organ donors were significantly older and their DRI was significantly increased. As donor-recipient sex-concordance was 82.7% for female recipients, they typically received organs with a significantly increased DRI. This might have affected the complication rate after transplantation while the significantly lower BMI of female recipients might have resulted in significantly shorter WIT for female recipients.

In the study by Wallis et al., male surgeons treated older patients and performed surgeries of greater complexity [1]. In our study cohort, donor and recipient baseline characteristics were similar in recipients transplanted by female and male surgeons; hence, transplant outcome with regards to surgeon sex is unlikely to be influenced by donor and recipient factors. Unfortunately, only 11.5% of female recipients in our study were treated by female surgeons. Positive effects of patient-surgeon sex concordance in liver transplantation might only become evident in larger study cohorts.

Yet, women still compose a small proportion of surgeons and have faced many training and professional challenges [23,24]. In the United States, although 50% of medical students and 40% of general surgery residents are female [25], less than 25% of doctors in ten surgical specialties are female [11]. Even fewer female surgeons are appointed to a leading position, with 7.4% chairs of surgery being occupied by a female [26]. In transplantation, similar to most surgical fields, female professionals remain underrepresented at all levels with only 13.1% of practicing transplant surgeons being female [27]. A recent study by the International Liver Transplantation Society (ILTS) collected data from 243 transplant centers worldwide and found that 18.2% of transplant surgeons and 31.9% of hepatologists were female. In the ILTS itself, 26.9% of the leadership positions were held by women [28].

Taking into consideration the increasing data reporting that sex-discordance leads to worse patient care and outcome, it is important to assess the impact of surgeon-patient sex discordance in surgery and especially in transplantation, a medical field clearly underrepresenting female expertise. Wallis et al. demonstrated that any patients treated by a female surgeon have a better 30-day survival rate, when considering both elective and emergency surgeries [2]. In a subgroup analysis including only emergency surgeries, surgeon sex alone did not influence 30-day patient survival. Transplantation in itself is an emergency surgical procedure; however, unlike other patients that require emergency surgery, most transplant patients have been ill for a long time and are familiar with the transplant center before surgery. Transplant recipients might even be acquainted with the center’s transplant surgeons.

In the Canadian cohort study, a discordance between surgeon and patient sex resulted in an increased rate of readmissions, and an increased 30-day morbidity and mortality. Especially for female patients, patient-surgeon sex discordance resulted in worse outcomes. Contrarily, male patients showed better outcomes when there was a discordance between patient and surgeon sex [1]. Choubey et al. investigated gender and race diversity among transplant department leaders for abdominal organs in the United States and its impact on patient outcomes [22]. Most transplant centers were led by a male surgical director (91.5%); 93.1% of the liver transplant programs had a male department leader, 91.2% of the kidney transplant programs had a male surgical director, and 90.3% of pancreas transplant programs were led by a male surgical director. Graft and patient survival of liver, kidney, and pancreas transplant programs were not influenced by leader sex. Direct personal interaction with the operating surgeons might be of greater importance. In our study, there was a trend for an improved five-year patient survival of female recipients treated by female surgeons; however, this did not reach statistical significance.

In patients with alcoholic cirrhosis, patient sex has also been reported to influence the outcomes after liver transplantation. Legaz and colleagues report that women with alcoholic cirrhosis showed less graft rejection than men with alcoholic cirrhosis [29]. They also report that women with hepatitis C had slightly lower survival rates than men in their cohort. The same group analyzed the influence of different clinical and sociodemographic causes on the short- and long-term outcomes after liver transplantation [30]. In the analyzed group, there were no differences in age and pre-transplant complications between female and male recipient. Furthermore, recipient sex did not influence post-transplant morbidity and mortality.

Multiple studies that looked into the influence of patient sex on the outcomes of liver transplantation report that after the introduction of the MELD score, women showed lower transplant rates and worse waitlist outcomes [31]. Women are more likely to become too ill to be transplantated and have a higher likelihood of death while waiting for an organ. Moreover, it appears that the differences in transplant rates are especially apparent in female patients with a MELD score over 15 [32]. Considering that the survival benefit with a liver transplant increases as MELD score increases, this means that female patients have a considerable loss in the survival benefit.

Donor sex has also been reported to have a significant influence on the outcomes of liver transplantation. Graft survival in patients receiving a donor-recipient gender match was reported to be superior to patients receiving a gender mismatch [33]. Male patients receiving an organ from a female donor appear to be the most affected group, with a significant lower graft survival compared with other donor-recipient gender groups. Not only in deceased donor transplantation, but also in living donor liver transplantation, grafts from female donors have a higher failure risk [34]. Donor sex showed a correlation with the posttransplant graft failure independent of the disparities in graft, vessels and bile duct size. Our study does not confirm this data. We found that both in female and male recipients five-year patient survival was not affected by recipient-donor sex-concordance.

In a recent study, Wallis and collegues have investigated whether differences in sex between the surgeon and anesthesiologist present at a surgery had an influence on the postoperative outcomes [35]. No association could be found between surgeon and anesthesiologist sex-discordance and the outcomes of the patients. Since liver transplantation represents a complex procedure that requires experienced multidisciplinary teams, it would be of interest to investigate if surgeon and anesthesiologist sex-discordance has an influence on liver transplant outcomes.

Our study has several limitations. The retrospective study design and selection bias due to the non-randomized study design could confound the results. Also, only biologic sex and not gender was accounted for in the analysis. Furthermore, the small size of the group of female recipients transplanted by a female surgeon might have not had sufficient power to evidence some findings. 

## 5. Conclusions

In conclusion, our study demonstrates that in liver transplantation both female recipients and female surgeons are underrepresented. Sex-discordance for transplant patients might influence long-term outcomes. To improve the care for all patients, larger studies need to investigate the influence of sex and gender on outcome of transplant recipients.

## Figures and Tables

**Figure 1 jpm-13-00281-f001:**
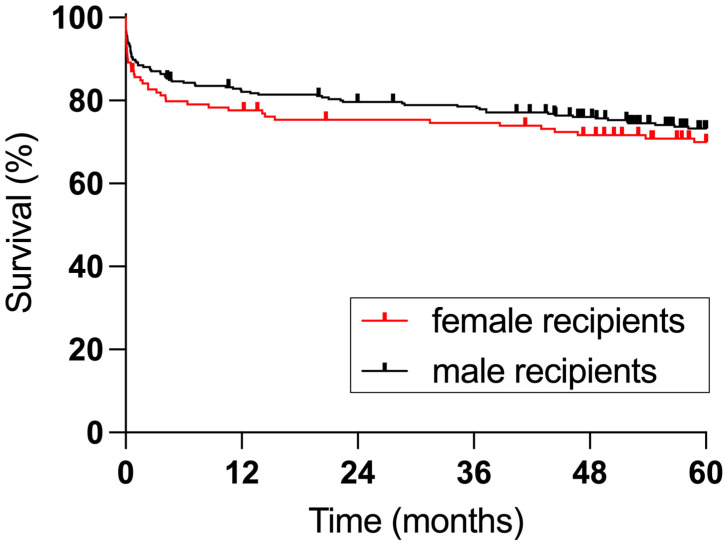
Survival of female and male recipients (*p* = 0.5665).

**Figure 2 jpm-13-00281-f002:**
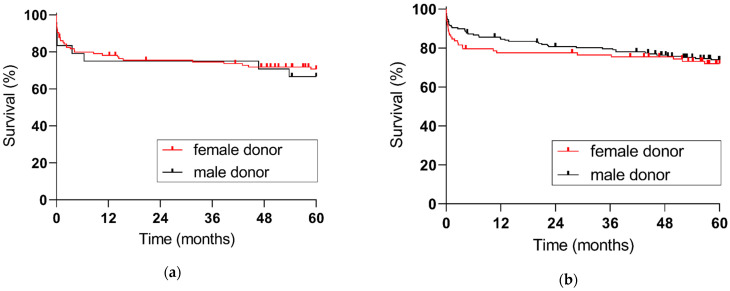
Five-year patient survival according to donor sex: (**a**) female recipients (*p* = 0.6909), (**b**) male recipients (*p* = 0.5882).

**Figure 3 jpm-13-00281-f003:**
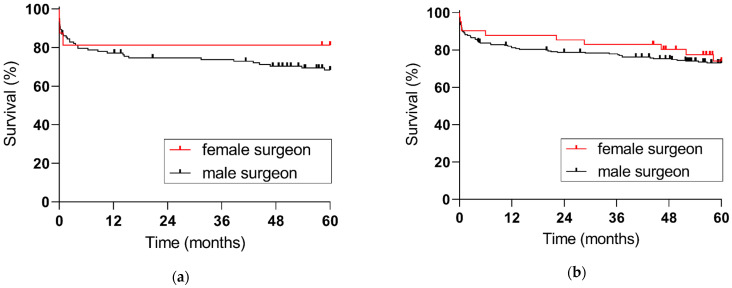
Five-year patient survival with regards to surgeon sex: (**a**) female recipients (*p* = 0.3621), (**b**) male recipients (*p* = 0.7238).

**Table 1 jpm-13-00281-t001:** Recipient, donor and surgical details stratified according to recipient sex.

	Female Recipients (*n* = 139)	Male Recipients (*n* = 286)	*p*
**Age (yrs)**	51 (18–69)	55 (18–71)	0.0594
**BMI (kg/m^2^)**	24.4 (15.1–41.5)	26.2 (16.1–44.3)	0.0009
**MELD score**	16 (6–40)	15 (6–40)	0.0716
**Diagnosis (%)**			<0.0001
**acute liver failure**	15.1	3.8	
**alcoholic cirrhosis**	17.3	21.0	
**hepatitis B/C**	5.8	14.0	
**HCC**	5.0	9.1	
**NASH**	15.1	25.5	
**PBC/PSC/AIH**	20.1	15.0	
**other**	21.6	11.5	
**Donor Risk Index**	1.837 (1.040–2.855)	1.746 (0.9730–2.678)	0.0272
**CIT (min)**	477 (226–1020)	470 (175–949)	0.5411
**WIT (min)**	28 (16–65)	30 (16–80)	0.0009

(body mass index, BMI; model of end-stage liver disease, MELD; hepatocellular carcinoma, HCC; non-alcoholic steatohepatitis, NASH; primary sclerosing cholangitis, PSC; primary biliary cholangitis, PBC; autoimmune hepatitis, AIH; cold ischemia time, CIT; warm ischemia time, WIT).

**Table 2 jpm-13-00281-t002:** Donor characteristics stratified according to donor sex.

	Female Donor (*n* = 213)	Male Donor (*n* = 212)	*p*
**Age (yrs)**	60 (1–89)	56 (1–87)	0.0182
**BMI (kg/m^2^)**	25.6 (12.5–51.4)	26.3 (13.3–58.8)	0.0012
**DRI**	1.908 (0.9730–2.678)	1.672 (0.9730–2.855)	<0.0001
**Brain death (%)**			0.0185
**Trauma**	7.98	13.21	
**Anoxia**	17.37	25.00	
**CVA**	69.01	54.25	
**other**	5.63	7.55	

(body mass index, BMI; donor risk index, DRI; cerebrovascular accident, CVA).

**Table 3 jpm-13-00281-t003:** Recipient, donor and surgical details stratified according to surgeon sex.

	Female Surgeon (*n* = 57)	Male Surgeon (*n* = 368)	*p*
**Recipient age**	54 (20–69)	54 (18–71)	0.5664
**Recipient BMI (kg/m^2^)**	26.5 (15.1–44.3)	25.4 (16.1–43.6)	0.4979
**Recipient MELD**	14 (7–40)	16 (6–40)	0.1452
**DRI**	1.837 (1.053–2.255)	1.768 (0.9730–2.855)	0.8372
**CIT (min)**	440 (323–787)	472 (175–1020)	0.3699
**WIT (min)**	34 (16–65)	29 (16–80)	0.0002

(body mass index, BMI; model of end-stage liver disease, MELD; donor risk index, DRI; cold ischemia time, CIT; warm ischemia time, WIT).

**Table 4 jpm-13-00281-t004:** Results from Cox regression analysis of factors independently associated with postoperative survival.

	Hazard Ratio (HR)	95% Confidence Interval (CI)	*p*
Recipient female sex	0.991	0.579, 1.697	0.975
Recipient age (per 10 yrs.)	1.204	1.017, 1.425	0.031
MELD Score	1.042	1.022, 1.063	<0.001
Donor Risk Index	1.311	0.797, 2.155	0.286
Surgeon-recipient sex concordance	1.074	0.638, 1.809	0.788

(model of end-stage liver disease, MELD).

## Data Availability

The data presented in this study are available on request from the corresponding author. The data are not publicly available due to ethical reasons.

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
