# Peer review of "Donor, Recipient and Surgeon Sex and Sex-Concordance and their Impact on Liver Transplant Outcome"

_jpm, 2023, doi:10.3390/jpm13020281_

Round 1

Reviewer 1 Report

Dear Editor,

the paper by Mazilescu et al. addresses a topic of current interest by investigating whether patient-donor-surgeon sex concordance influences short- and long-term outcomes of liver transplantation. This single-center retrospective study is acceptably conducted and described in fluent English language. However, it does not provide meaningful results which also lack interpretability, due to the sample size and the 'exploratory' nature of the statistical data.

Besides these general observations, other points can be raised:
1) In the Author section, please indicate with the same number the same Affiliation of the Authors.
2) The discussion of the study should be more focused in the interpretation of the results of the study itself, therefore several notions reported in the Discussion section should be stated in the Introduction paragraph (which is also rather brief).
3) A description of gender inequality in the transplant waiting list could also be added to the Introduction. In fact, it is well known that female cirrhotic patients are disadvantaged by the MELD-score and new models (e.g., MELD 3.0) are being applied to eliminate this gap.
4) The characteristics of immunosuppressive therapy reported in the "Materials and Methods" section need to be considered as “Results”.
5) In Table 1 it is necessary to a) report more information on the recipients(how many patients have cardiovascular diseases, which are the main cause of early adverse events), b) separate HCC (whose percentages are surprisingly low) from the other diagnoses and indicate for each the respective p-value between the two groups; c) indicate how many patients received organs from higher risk donors (e.g., DCD).
6) Line 86-87: this concept needs to be more stressed and explained in the conclusions as it results in a lack of the paper.
7) Line 137: please explain why the graft loss occurred.
8) Line 142: please report survival rates at 1 and 5 years post-transplantation.
9) Line 152: Looking at Fig. 3 it appears that even in female recipients 5-year survival was not affected by patient-surgeon sex concordance, please clarify.

Author Response

Dear Editor,

the paper by Mazilescu et al. addresses a topic of current interest by investigating whether patient-donor-surgeon sex concordance influences short- and long-term outcomes of liver transplantation. This single-center retrospective study is acceptably conducted and described in fluent English language. However, it does not provide meaningful results which also lack interpretability, due to the sample size and the 'exploratory' nature of the statistical data.

We thank the reviewer for these valuable comments and the opportunity to resubmit a revised version of the manuscript. Please find below the point-by-point responses to the comments of the reviewer.

Besides these general observations, other points can be raised:

  • In the Author section, please indicate with the same number the same Affiliation of the Authors.

We thank the reviewer for this suggestion. We have indicated the Affiliations according to the journal’s template. Since the email addresses of the authors are different, each of them has received a different number.

  • The discussion of the study should be more focused in the interpretation of the results of the study itself, therefore several notions reported in the Discussion section should be stated in the Introduction paragraph (which is also rather brief).

We thank the reviewer for these helpful comments to improve the manuscript. We have now added a paragraph regarding the aspects pointed out above in the “Introduction” section. Please see page 2 of the revised manuscript.

  • A description of gender inequality in the transplant waiting list could also be added to the Introduction. In fact, it is well known that female cirrhotic patients are disadvantaged by the MELD-score and new models (e.g., MELD 3.0) are being applied to eliminate this gap.

We thank the reviewer for pointing out this important matter. We have now added a paragraph regarding this topic in the “Introduction” section. Please see page 2 of the revised manuscript.

  • The characteristics of immunosuppressive therapy reported in the "Materials and Methods" section need to be considered as “Results”.

We thank the reviewer for this observation. Since immunosuppressive therapy is applied in a standardized matter, regardless of the study that we are conducting, we believe that it should be reported in the “Materials and Methods” section.

  • In Table 1 it is necessary to a) report more information on the recipients(how many patients have cardiovascular diseases, which are the main cause of early adverse events), b) separate HCC (whose percentages are surprisingly low) from the other diagnoses and indicate for each the respective p-value between the two groups; c) indicate how many patients received organs from higher risk donors (e.g., DCD).

We thank the reviewer for these suggestions. a) At our center, recipients are extensively screened for cardiovascular diseases (echocardiography, cardiac computer tomography, coronary angiography) and associated adverse events. Hence, we do not share the experience of cardiovascular diseases being the main cause of early adverse events. b) AS you pointed out yourself, HCC is not an extremely frequent diagnosis leading to transplantation in our cohort. Therefore, it is included within all diagnoses in table 1. To demonstrate the difference in diagnoses according to recipient sex in total, we analyzed all diagnoses for differences between groups together. c) All patients in our study received an organ from donation after brain death. Donation after cardiac death is not yet allowed in Germany. We have included a comment in this regard in the “Materials and Methods” section, please see page 2 of the revised manuscript.

  • Line 86-87: this concept needs to be more stressed and explained in the conclusions as it results in a lack of the paper.

Unfortunately, we did not understand what the reviewer meant with this comment. We kindly ask the reviewer to rephrase this comment.

  • Line 137: please explain why the graft loss occurred.

We thank the reviewer for pointing out this important matter. We have now added a paragraph regarding this aspect in the “Results” section. Please see page 5 of the revised manuscript.

  • Line 142: please report survival rates at 1 and 5 years post-transplantation.

We thank the reviewer for this question. We have now added the survival rates at 1- and 5-years post-transplantation. Please see page 5 of the revised manuscript.

9) Line 152: Looking at Fig. 3 it appears that even in female recipients 5-year survival was not affected by patient-surgeon sex concordance, please clarify.

We thank the reviewer for this observation. Indeed, in our cohort, survival in female recipients was not affected by patient-surgeon sex concordance. Female recipients treated by female surgeons did show an improved five-year patient survival, however this did not reach statistical significance. 

Reviewer 2 Report

The article by Mazilescu et al. is well written and structured and could be publishable in the journal. However, I will further discuss some points and shortcomings that in my opinion do not appear in the revised manuscript. First, it seems as if being a female surgeon is more important than being a male surgeon for the best evolution of liver transplantation. In this sense, I would carry out a statistical analysis of confounding factors, comparing which factors are really key and which may be biasing others. The Cox regression analysis is missing the sex of surgeon.

The underlying pathology that led to the transplant is also very different in men than in women in their study, where acute liver failure or autoimmune and inflammatory pathologies are much more frequent in women than in men. The same is true on the contrary, HBV/HCV is much more frequent in men than women and NASH.
In Table 1, PBC/PSC/AIH is included as a diagnosis and I do not understand very well how the authors include two PBC/PSC pathologies that lead to transplantation with the cold ischemia time, which is a parameter of organ donation. They must go separate.
Lastly, there is a lack of important publications regarding sex in liver transplantation in the indicated references, and that should be added, such as:

Bolarin et al. Causes of Death and Survival in Alcoholic Cirrhosis Patients Undergoing Liver Transplantation: Influence of the Patient's Clinical Variables and Transplant Outcome Complications. Diagnostics (Basel). 2021;11(6):968. doi: 10.3390/diagnostics11060968.

 Legaz I, et al. Patient Sex in the Setting of Liver Transplant in Alcoholic Liver Disease. Exp Clin Transplant. 2019 Jun;17(3):355-362. doi: 10.6002/ect.2017.0302.

Author Response

The article by Mazilescu et al. is well written and structured and could be publishable in the journal. However, I will further discuss some points and shortcomings that in my opinion do not appear in the revised manuscript.

We thank the reviewer for pointing out several important matters and the opportunity to resubmit a revised version of the manuscript. Please find below the point-by-point responses to the comments of the reviewer.

First, it seems as if being a female surgeon is more important than being a male surgeon for the best evolution of liver transplantation. In this sense, I would carry out a statistical analysis of confounding factors, comparing which factors are really key and which may be biasing others. The Cox regression analysis is missing the sex of surgeon.

We thank the reviewer for pointing out this important issue. Transplant waitlist patients suffer from a similar, life-threatening disease and undergo the same procedure; therefore, additional confounding factors are limited in this cohort. Since we included surgeon-recipient sex concordance in the Cox regression analysis, we did not include the surgeon sex in the analysis.

The underlying pathology that led to the transplant is also very different in men than in women in their study, where acute liver failure or autoimmune and inflammatory pathologies are much more frequent in women than in men. The same is true on the contrary, HBV/HCV is much more frequent in men than women and NASH. In Table 1, PBC/PSC/AIH is included as a diagnosis and I do not understand very well how the authors include two PBC/PSC pathologies that lead to transplantation with the cold ischemia time, which is a parameter of organ donation. They must go separate.

We thank the reviewer for this question. In Table 1 we included recipient, donor and surgical details separated by recipient sex. Because all parameters are relevant for the recipient, we decided to include recipient diagnosis, cold ischemia time and DRI in the same table. No correlation or analysis was presented in this table. Immunological disorders leading to cirrhosis and transplantation (PBC/PSC/AHI) were presented together in the table as a combined subheading for better readability. 

Lastly, there is a lack of important publications regarding sex in liver transplantation in the indicated references, and that should be added, such as:

Bolarin et al. Causes of Death and Survival in Alcoholic Cirrhosis Patients Undergoing Liver Transplantation: Influence of the Patient's Clinical Variables and Transplant Outcome Complications. Diagnostics (Basel). 2021;11(6):968. doi: 10.3390/diagnostics11060968.

 Legaz I, et al. Patient Sex in the Setting of Liver Transplant in Alcoholic Liver Disease. Exp Clin Transplant. 2019 Jun;17(3):355-362. doi: 10.6002/ect.2017.0302.

We thank the reviewer for these helpful comments to improve the manuscript. We have now added a paragraph regarding the papers mentioned above in the “Discussion” section. Please see page 8 of the revised manuscript.

Round 2

Reviewer 1 Report

Dear Editor,

the Authors have clarified the questions I raised.

Some minor spell check are still required (e.g., "female" instead of "feamle", line 170).